# Learn Locally, Correct Globally: A Distributed Algorithm for Training Graph Neural Networks

**Morteza Ramezani**[∗], **Weilin Cong**[∗], **Mehrdad Mahdavi**
**Mahmut T. Kandemir**, **Anand Sivasubramaniam**
Pennsylvania State University, University Park, PA 16802, USA
`morteza@cse.psu.edu, wxc272@psu.edu, mzm616@psu.edu`
`anand@cse.psu.edu, kandemir@cse.psu.edu`

## Abstract

Despite the recent success of Graph Neural Networks (GNNs), training GNNs on large graphs remains challenging. The limited resource capacities of the existing servers, the dependency between nodes in a graph, and the privacy concern due to the centralized storage and model learning have spurred the need to design an effective distributed algorithm for GNN training. However, existing distributed GNN training methods impose either excessive communication costs or large memory overheads that hinders their scalability. To overcome these issues, we propose a communication-efficient distributed GNN training technique named *Learn Locally, Correct Globally* (LLCG). To reduce the communication and memory overhead, each local machine in LLCG first trains a GNN on its local data by ignoring the dependency between nodes among different machines, then sends the locally trained model to the server for periodic model averaging. However, ignoring node dependency could result in significant performance degradation. To solve the performance degradation, we propose to apply *Global Server Corrections* on the server to refine the locally learned models. We rigorously analyze the convergence of distributed methods with periodic model averaging for training GNNs and show that naively applying periodic model averaging but ignoring the dependency between nodes will suffer from an irreducible residual error. However, this residual error can be eliminated by utilizing the proposed global corrections to entail fast convergence rate. Extensive experiments on real-world datasets show that LLCG can significantly improve the efficiency without hurting the performance.

## 1 Introduction

In recent years, Graph Neural Networks (GNNs) have achieved impressive results across numerous graph-based applications, including social networks (Hamilton et al., 2017; Deng et al., 2019), recommendation systems (Ying et al., 2018; Wang et al., 2018), and drug discovery (Fout et al., 2017; Do et al., 2019; Ghorbani et al., 2022; Faez et al., 2021). Despite their recent success, effective training of GNNs on large-scale real-world graphs, such as Facebook social network (Boldi & Vigna, 2004), remains challenging. Although several attempts have been made to scale GNN training by sampling techniques (Hamilton et al., 2017; Zou et al., 2019; Zeng et al., 2020; Chiang et al., 2019; Chen et al., 2018; Zhang et al., 2021; Ramezani et al., 2020), they are still inefficient for training on extremely large graphs, due to the unique structure of GNNs and the limited memory capacity/bandwidth of current servers. One potential solution to tackle these limitations is employing distributed training with data parallelism, which have become almost a *de facto* standard for fast and accurate training for natural language processing (Lin et al., 2021; Hard et al.,

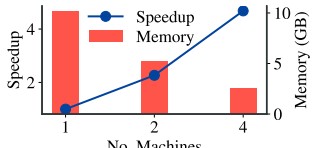

Figure 1: Comparison of the speedup and the memory consumption of distributed multi-machine training and centralized single machine training on the `Reddit` dataset.

---

[∗]Equal Contribution.

2018) and computer vision (Bonawitz et al., 2019; Konečný et al., 2018). For example, as shown in Figure 1, moving from single machine to multiple machines reduces the training time and alleviates the memory burden on each machine. Besides, scaling the training of GNNs with sampling techniques can result in privacy concerns: existing sampling-based methods require centralized data storage and model learning, which could result in privacy concerns in real-world scenarios (Shin et al., 2018; Wu et al., 2021). Fortunately, the privacy in distributed learning can be preserved by avoiding mutual access to data between different local machines, and using only a trusted third party server to access the entire data.

Nonetheless, generalizing the existing data parallelism techniques of classical distributed training to the graph domain is non-trivial, which is mainly due to the dependency between nodes in a graph. For example, unlike solving image classification problems where images are mutually independent, such that we can divide the image dataset into several partitions without worrying about the dependency between images; GNNs are heavily relying on the information inherent to a node and its neighboring nodes. As a result, partitioning the graph leads to subgraphs with edges spanning subgraphs (*cut-edges*), which will cause information loss and hinder the performance of the model (Angerd et al., 2020). To cope with this problem, (Md et al., 2021; Jiang & Rumi, 2021; Angerd et al., 2020) propose to transfer node features and (Zheng et al., 2020; Tripathy et al., 2020; Scardapane et al., 2020) propose to transfer both the node feature and its hidden embeddings between local machines, both of which can cause significant storage/communication overhead and privacy concerns (Shin et al., 2018; Wu et al., 2021).

To better understand the challenge of distributed GNN training, we compare the validation F1-score in Figure 2 (a) and the average data communicated per round in Figure 2 (b) for two different distributed GNN training methods on the `Reddit` dataset. On the one hand, we can observe that when ignoring the cut-edges, *Parallel SGD with Periodic Averaging* (PSGD-PA (Dean et al., 2012; Li et al., 2020b)) suffers from significant accuracy drop and cannot achieve the same accuracy as the single machine training, even by increasing the number of communication. However, *Global Graph Sampling* (GGS) can successfully reach the baseline by considering the cut-edges and allowing feature transfer, at the cost of significant communication overhead, and potential violation of privacy.

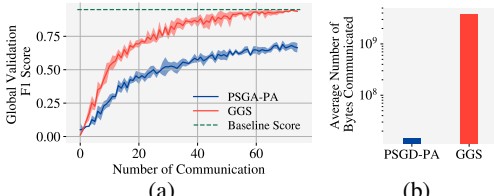

Figure 2: Comparison of (a) the validation F1-score and (b) the average data communicated per round (in bytes and log-scale) for two different distributed GNN training settings, including *Parallel SGD with Periodic Averaging* (PSGD-PA) where the cut-edges are ignored and only the model parameters are transferred and *Global Graph Sampling* (GGS), where the cut-edges are considered and the node features of the cut-edges are transferred to the corresponding local machine, on the `Reddit` dataset using 8 machines.

In this paper, we propose a communication-efficient distributed GNN training method, called *Learn Locally, Correct Globally* (LLCG). To reduce the communication overhead, inspired by the recent success of the distributed optimization with periodic averaging (Stich, 2019; Yu et al., 2019), we propose *Local Training with Periodic Averaging*: where each local machine first locally trains a GNN model by ignoring the cut-edges, then sends the trained model to the server for periodic model averaging, and receive the averaged model from server to continue the training. By doing so we eliminate the features exchange phase between server and local machines, but it can result in a significant performance degradation due to the lack of the global graph structure and the dependency between nodes among different machines. To compensate for this error, we propose a *Global Server Correction* scheme to take advantage of the available global graph structure on the server and refine the averaged locally learned models before sending it back to each local machine. Notice that without *Global Server Correction*, LLCG is similar to PSGD-PA as introduced in Figure 2.

To get a deeper understanding on the necessity of *Global Server Correction*, we provide the first theoretical analysis on the convergence of distributed training for GNNs with periodic averaging. In particular, we show that solely averaging the local machine models and ignoring the global graph structure will suffer from an irreducible residual error, which provides sufficient explanation on why *Parallel SGD with Periodic Averaging* can never achieve the same performance as the model trained on a single machine in Figure 2 (a). Then, we theoretically analyze the convergence of our proposal LLCG . We show that by carefully choosing the number of global correction steps,

LLCG can overcome the aforementioned residual error and enjoys $\mathcal{O}\big(1/\sqrt{PT}\big)$ convergence rate with $P$ local machines and $T$ iterations of gradient updates, which matches the rate of (Yu et al., 2019) on a general (not specific for GNN training) non-convex optimization setting. Finally, we conduct comprehensive evaluations on real-world graph datasets with ablation study to validate the effectiveness of LLCG and its improvements over the existing distributed methods.

**Related works.** Recently, several attempts have been made on distributed GNN training. According to how they deal with the input/hidden feature of nodes that are associated with the cut-edges (i.e., the edges spanning subgraphs of each local machine), existing methods can be classified into two main categories: (1) *Input feature only communication-based methods*: In these methods, each local machine receives the input features of all nodes required for the gradient computation from other machines, and trains individually. However, since the number of required nodes grows exponentially with the number of layers, these methods suffer from a significant communication and storage overhead. To alleviate these issues, (Md et al., 2021) proposes to split the original graph using a min-cut graph partition algorithm that can minimize the number of cut-edges. (Jiang & Rumi, 2021) proposes to use importance sampling to assign nodes on the local machine with a higher probability. (Angerd et al., 2020) proposes to sample and save a small subgraph from other local machines as an approximation of the original graph structure. Nonetheless, these methods are limited to a very shallow GNN structure and suffer from significant performance degradation when the original graph is dense. (2) *Input and hidden feature communication-based methods*: These methods propose to communicate hidden features in addition to the input node features. Although these methods reduce the number of transferred bytes during each communication round (due to the smaller size of hidden embedding and less required nodes features), the number of communication rounds grows linearly as the number of layers, and are prone to more communication delay. To address these issues, in addition to optimal partitioning of the graph, (Zheng et al., 2020) proposes to use sparse embedding to reduce the number of bytes to communicate and (Tripathy et al., 2020) proposes several graph partitioning techniques to diminish the communication overhead.

## 2 BACKGROUND AND PROBLEM FORMULATION

In this section, we start by describing Graph Convolutional Network (GCN) and its training algorithm on a single machine, then formulate the problem of distributed GCN training. Note that we use GCN with mean aggregation for simplicity, however, our discussion is also applicable to other GNN architectures, such as SAGE (Hamilton et al., 2017), GAT (Velickovic et al., 2018), ResGCN (Li et al., 2019) and APPNP (Klicpera et al., 2019).

**Training GCN on a single machine.** Here, we consider the semi-supervised node classification in an undirected graph $\mathcal{G}(\mathcal{V}, \mathcal{E})$ with $N = |\mathcal{V}|$ nodes and $|\mathcal{E}|$ edges. Each node $v_i \in \mathcal{V}$ is associated with a pair $(\mathbf{x}_i, \mathbf{y}_i)$, where $\mathbf{x}_i \in \mathbb{R}^d$ is the input feature vector, $\mathbf{y}_i \in \mathbb{R}^{|\mathcal{C}|}$ is the ground truth label, and $\mathcal{C}$ is the candidate labels in the multi-class classifications. Besides, let $\mathbf{X} = [\mathbf{x}_1, \ldots, \mathbf{x}_N] \in \mathbb{R}^{N \times d}$ denote the input node feature matrix. Our goal is to find a set of parameters $\boldsymbol{\theta} = \{\mathbf{W}^{(\ell)}\}_{\ell=1}^{L}$ by minimizing the empirical loss $\mathcal{L}(\boldsymbol{\theta})$ over all nodes in the training set, i.e.,

$$\mathcal{L}(\boldsymbol{\theta}) = \frac{1}{N} \sum_{i \in \mathcal{V}} \phi(\mathbf{h}_i^{(L)}, \mathbf{y}_i), \qquad \mathbf{h}_i^{(\ell)} = \sigma\Big(\frac{1}{|\mathcal{N}(v_i)|} \sum_{j \in \mathcal{N}(v_i)} \mathbf{h}_j^{(\ell-1)} \mathbf{W}^{(\ell)}\Big), \qquad (1)$$

where $\phi(\cdot, \cdot)$ is the loss function (e.g., cross entropy loss), $\sigma(\cdot)$ is the activation function (e.g., ReLU), and $\mathcal{N}(v_i)$ is the neighborhood of node $v_i$. In practice, we can update the model parameters by the stochastic gradient computed on a sampled mini-batch (using full-neighbors) by

$$\tilde{\nabla}\mathcal{L}(\boldsymbol{\theta}, \xi) = \frac{1}{B} \sum_{i \in \xi} \nabla\phi(\mathbf{h}_i^{(L)}, \mathbf{y}_i), \qquad (2)$$

where $\xi$ denotes an i.i.d. sampled mini-batch of size $B$ and we have $\mathbb{E}[\tilde{\nabla}\mathcal{L}(\boldsymbol{\theta}, \xi)] = \nabla\mathcal{L}(\boldsymbol{\theta})$.

**Distributed GCN training with periodic averaging.** In this paper, we consider the distributed learning setting with $P$ local machines and a single parameter server. The original input graph $\mathcal{G}$ is partitioned into $P$ subgraphs, where $\mathcal{G}_p(\mathcal{V}_p, \mathcal{E}_p)$ denotes the subgraph on the $p$-th local machine with $N_p = |\mathcal{V}_p|$ nodes, and $\mathbf{X}_p \in \mathbb{R}^{N_p \times d}$ as the input feature of all nodes in $\mathcal{V}_p$ located on the $p$-th machine. Then, the full-batch local gradient $\nabla\mathcal{L}_p^{\text{local}}(\boldsymbol{\theta}_p)$ is computed as

$$\nabla\mathcal{L}_p^{\text{local}}(\boldsymbol{\theta}_p) = \frac{1}{N_p} \sum_{i \in \mathcal{V}_p} \nabla\phi(\mathbf{h}_i^{(L)}, \mathbf{y}_i), \quad \mathbf{h}_i^{(\ell)} = \sigma\Big(\frac{1}{|\mathcal{N}_p(v_i)|} \sum_{j \in \mathcal{N}_p(v_i)} \mathbf{h}_j^{(\ell-1)} \mathbf{W}_p^{(\ell)}\Big), \quad (3)$$

---

**Algorithm 1** Distributed GCN training with "*Parallel SGD with Periodic Averaging*"

---

**Input:** Global parameters $\bar{\boldsymbol{\theta}}^0$, local parameters $\boldsymbol{\theta}_p^0 = \bar{\boldsymbol{\theta}}^0$, time-step $t = 0$, learning rate $\eta$.
1: **for** $r \leftarrow 1$ **to** $R$ **do**
2:     **for** $p \leftarrow 1$ **to** $P$ **do in parallel**           ▷ Parallel training on local machines
3:         Local machine $p$ receives the global parameters $\boldsymbol{\theta}_p^t \leftarrow \bar{\boldsymbol{\theta}}^t$.     ▷ Communication
4:         **for** $k \leftarrow 1$ **to** $K$ **do**
5:             $t \leftarrow t + 1$.
6:             Local machine $p$ constructs the mini-batch $\xi_p^t$ with neighbor sampling.
7:             Local machine $p$ computes the stochastic gradients $\tilde{\nabla}\mathcal{L}_p^{\text{local}}(\boldsymbol{\theta}_p^t, \xi_p^t)$.
8:             Local machine $p$ updates the local parameter by $\boldsymbol{\theta}_p^{t+1} = \boldsymbol{\theta}_p^t - \eta\tilde{\nabla}\mathcal{L}_p^{\text{local}}(\boldsymbol{\theta}_p^t, \xi_p^t)$.
9:         **end for**
10:        Local machine $p$ sends the local parameters $\boldsymbol{\theta}_p^{t+1}$ to the server. ▷ Communication
11:     **end for**
12:     Server updates the global parameters by parameter averaging $\bar{\boldsymbol{\theta}}^{t+1} = \frac{1}{P}\sum_{p=1}^P \boldsymbol{\theta}_p^{t+1}$.
13: **end for**
**Output:** Server returns trained GCN model with $\min_t \mathbb{E}[\|\nabla\mathcal{L}(\bar{\boldsymbol{\theta}}^t)\|^2]$.

---

where $\boldsymbol{\theta}_p = \{\mathbf{W}_p^{(\ell)}\}_{\ell=1}^L$ is the model parameters on the $p$-th local machine, $\mathcal{N}_p(v_i) = \{v_j | (v_i, v_j) \in \mathcal{E}_p\}$ is the local neighbors of node $v_i$ on the $p$-th local machine. When the graph is large, the computational complexity of forward and backward propagation could be very high. One practical solution is to compute the stochastic gradient on a sampled mini-batch with neighbor sampling, i.e.,

$$\tilde{\nabla}\mathcal{L}_p^{\text{local}}(\boldsymbol{\theta}_p, \xi_p) = \frac{1}{B_p}\sum_{i \in \xi_p} \nabla\phi(\tilde{\mathbf{h}}_i^{(L)}, \mathbf{y}_i), \ \tilde{\mathbf{h}}_i^{(\ell)} = \sigma\Big(\frac{1}{|\tilde{\mathcal{N}}_p(v_i)|}\sum_{j \in \tilde{\mathcal{N}}_p(v_i)} \tilde{\mathbf{h}}_j^{(\ell-1)}\mathbf{W}_p^{(\ell)}\Big), \quad (4)$$

where $\xi_p$ is an i.i.d. sampled mini-batch of $B_p$ nodes, $\tilde{\mathcal{N}}_p(v_i) \subset \mathcal{N}(v_i)$ is the sampled neighbors.

An illustration of distributed GCN training with *Parallel SGD with Periodic Averaging* (PSGD-PA) is summarized in Algorithm 1. Before training, the server maintains a global model $\bar{\boldsymbol{\theta}}^0$ and each local machine keeps a local copy of the same model $\boldsymbol{\theta}_p^0$. During training, the local machine first updates the local model $\boldsymbol{\theta}_p^t$ using the stochastic gradient $\tilde{\nabla}\mathcal{L}_p^{\text{local}}(\boldsymbol{\theta}_p^t, \xi_p^t)$ computed by Eq. 4 for $K$ iterations (line 8), then sends the local model $\boldsymbol{\theta}_p^t$ to the server (line 10). At each communication step, the server collects and averages the model parameters from the local machines (line 12) and send the averaged model $\boldsymbol{\theta}_p^{t+1}$ back to each local machine.

**Limitations.** Although PSGD-PA can significantly reduce the communication overhead by transferring the locally trained models instead of node feature/embeddings (refer to Figure 2 (b)), it suffers from performance degeneration due to ignorance of the cut-edges (refer to Figure 2 (a)). In the next section, we introduce a communication-efficient algorithm LLCG that does not suffer from this issue, and can achieve almost the same performance as training the model on a single machine.

## 3    PROPOSED ALGORITHM: LEARN LOCALLY CORRECT GLOBALLY

In this section, we describe *Learn Locally, Correct Globally* (LLCG) for distributed GNN training. LLCG includes two main phases, *local training with periodic model averaging* and *global server correction*, to help reduce both the number of required communications and size of transferred data, without compromising the predictive accuracy. We summarize the details of LLCG in Algorithm 2.

### 3.1    LOCAL TRAINING WITH PERIODIC MODEL AVERAGING

At the beginning of a local epoch, each local machine receives the latest global model parameters from the server (line 3). Next, each local machine runs $K\rho^r$ iterations to update the local model (line 4 to 9), where $K$ and $\rho$ are the hyper-parameters that control the local epoch size. Note that instead of using a fixed local epoch size as Algorithm 1, we choose to use exponentially increasing local epoch size in LLCG with $\rho > 1$. The reasons are as follows.

At the beginning of the training phase, all local models $\boldsymbol{\theta}_p^t$ are far from the optimal solution and will receive a gradient $\tilde{\nabla}\mathcal{L}_p^{\text{local}}(\boldsymbol{\theta}_p^t, \xi_p^t)$ computed by Eq. 4. Using a smaller local update step at the early stage guarantees each local model does not diverge too much from each other before the model averaging step at the server side (line 12). However, towards the end of the training, all local

---

**Algorithm 2** Distributed GCN training by "*Learn Locally, Correct Globally*"

---

**Input:** Global parameters $\bar{\boldsymbol{\theta}}^0$, local parameters $\boldsymbol{\theta}_p^0$, time-step $t = 0$, local step size hyper-parameters $K, \rho$, and learning rate $\gamma, \eta$

1: **for** $r \leftarrow 1$ **to** $R$ **do**
2:     **for** $p \leftarrow 1$ **to** $P$ **do in parallel**                          ▷ Parallel training on local machine
3:         Local machine $p$ receives the global parameters $\boldsymbol{\theta}_p^t \leftarrow \bar{\boldsymbol{\theta}}^t$         ▷ Communication
4:         **for** $k \leftarrow 1$ **to** $K\rho^r$ **do**
5:             $t \leftarrow t + 1$
6:             Local machine $p$ constructs the mini-batch $\xi_p^t$ with neighbor sampling
7:             Local machine $p$ computes stochastic gradients $\tilde{\nabla}\mathcal{L}_p^{\text{local}}(\boldsymbol{\theta}_p^t, \xi_p^t)$
8:             Local machine $p$ updates model parameter by $\boldsymbol{\theta}_p^{t+1} = \boldsymbol{\theta}_p^t - \eta\tilde{\nabla}\mathcal{L}_p^{\text{local}}(\boldsymbol{\theta}_p^t, \xi_p^t)$
9:         **end for**
10:        Local machine $p$ sends the local parameters $\boldsymbol{\theta}_p^{t+1}$ to the server   ▷ Communication
11:    **end for**
12:    Server updates the global parameters using parameter averaging $\bar{\boldsymbol{\theta}}^{t+1} = \frac{1}{P}\sum_{p=1}^P \boldsymbol{\theta}_p^{t+1}$
13:    **for** $s \leftarrow 1$ **to** $S$ **do**                                          ▷ Server Correction
14:        $t \leftarrow t + 1$
15:        Server constructs a mini-batch $\xi^t$ with full-neighbors
16:        Server computes the stochastic gradient $\tilde{\nabla}\mathcal{L}(\bar{\boldsymbol{\theta}}^t, \xi^t)$
17:        Server updates the global parameters by $\bar{\boldsymbol{\theta}}^{t+1} = \bar{\boldsymbol{\theta}}^t - \gamma\tilde{\nabla}\mathcal{L}(\bar{\boldsymbol{\theta}}^t, \xi^t)$
18:    **end for**
19: **end for**
**Output:** Server return GCN model with trained $\min_t \mathbb{E}[\|\nabla\mathcal{L}(\bar{\boldsymbol{\theta}}^t)\|^2]$

---

models $\boldsymbol{\theta}_p^t$ will receive relatively smaller gradient $\tilde{\nabla}\mathcal{L}_p^{\text{local}}(\boldsymbol{\theta}_p^t, \xi_p^t)$, such that we can chose a larger local epoch size to reduce the number of communications, without worrying about the divergence of local models. By doing so, after total number of $T = \sum_{r=1}^R K\rho^r$ iterations, LLCG only requires $R = \log_\rho \frac{T}{K}$ rounds of communications. Therefore, compared to the fully-synchronous method, we can significantly reduce the total number of communications from $\mathcal{O}(T)$ to $\mathcal{O}(\log_\rho \frac{T}{K})$.

## 3.2 GLOBAL SERVER CORRECTION

The design of the global server correction is to ensure that the trained model not only learns from the data on each local machine, but also learns the global structure of the graph, thus reducing the information loss caused by graph partitioning and avoiding cut-edges. Before the correction, the server receives the locally trained models from all local machines (line 10) and applies model parameter averaging (line 12). Next, $S$ server correction steps are applied on top of the averaged model (line 13 to 18). During the correction, the server first constructs a mini-batch $\xi^t$ using full-neighbors[1] (line 15), compute the stochastic gradient $\tilde{\nabla}\mathcal{L}(\bar{\boldsymbol{\theta}}^t, \xi^t)$ on the constructed mini-batch by Eq. 2 (line 16) and update the averaged model $\bar{\boldsymbol{\theta}}^t$ for $S$ iterations (line 17). The number of correction steps $S$ [2] depends on the heterogeneity among the subgraphs on each local machine: the more heterogeneous the subgraphs are, the more correction steps are required to better refine the averaged model and reduce the divergence across the local models. Note that, the heterogeneity is minimized when employing GGS (Figure 2) with the local machines having access to the full graph, as a result. However, GGS requires sampling from the global graph and communication at every iteration, which results in additional overhead and lower efficiency. Instead, in LLCG we are trading computation on the server for the costly feature communication, and only requires periodic communication.

## 4 THEORETICAL ANALYSIS

In this section, we provide the convergence analysis on the distributed training of GCN under two different settings, i.e., with and without server correction. We first introduce the notations and assumptions for the analysis (Section 4.1). Then, we show that periodic averaging of local machine models alone and ignoring the global graph structure will suffer from an irreducible residual error (Section 4.2). Finally, we show that this residual error can be eliminated by running server correction steps after each periodic averaging step on the server (Section 4.3).

---

[1] Note that using full neighbors is required for the server correction but not the local machines
[2] In practice, we found $S = 1$ or $S = 2$ works well on most datasets.

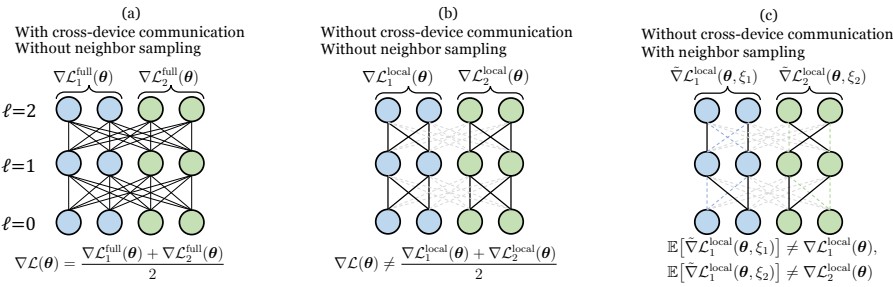

Figure 3: Comparison of notations $\nabla\mathcal{L}_p^{\text{local}}(\boldsymbol{\theta})$, $\tilde{\nabla}\mathcal{L}_p^{\text{local}}(\boldsymbol{\theta},\xi_p)$, and $\nabla\mathcal{L}_p^{\text{full}}(\boldsymbol{\theta})$ on two local machines, where the blue node and green circles represent nodes on different local machines.

### 4.1 NOTATIONS AND ASSUMPTIONS

Let us first recall the notations defined in Section 2, where $\mathcal{L}(\boldsymbol{\theta})$ denotes the global objective function computed using the all node features $\mathbf{X}$ and the original graph $\mathcal{G}$, $\mathcal{L}_p(\boldsymbol{\theta})$ denotes the local objective function computed using the local node features $\mathbf{X}_p$ and local graph $\mathcal{G}_p$, $\boldsymbol{\theta}_p^t$ denotes the model parameters on the $p$-th local machine at the $t$-th step, and $\bar{\boldsymbol{\theta}}^t = \frac{1}{P}\sum_{p=1}^P \boldsymbol{\theta}_p^t$ denotes the virtual averaged model at the $t$-th step. In the non-convex optimization, our goal is to show the expected gradient of the global objective on the virtual averaged model parameters $\mathbb{E}[\|\nabla\mathcal{L}(\bar{\boldsymbol{\theta}}^t)\|^2]$ decreases as the number of local machines $P$ and the number of training steps $T$ increase. Besides, we introduce $\nabla\mathcal{L}_p^{\text{full}}(\boldsymbol{\theta})$ as the gradient computed on the $p$-th local machine but have access the full node features $\mathbf{X}$ and the original graph structure $\mathcal{G}$ as

$$\nabla\mathcal{L}_p^{\text{full}}(\boldsymbol{\theta}) = \frac{1}{|\mathcal{V}_p|}\sum_{i\in\mathcal{V}_p}\nabla\phi(\mathbf{h}_i^{(L)}, y_i), \qquad \mathbf{h}_i^{(\ell)} = \sigma\Big(\frac{1}{|\mathcal{N}(v_i)|}\sum_{j\in\mathcal{N}(v_i)}\mathbf{h}_j^{(\ell-1)}\mathbf{W}_p^{(\ell)}\Big). \quad (5)$$

Please refer to Figure 3 for an illustration of different gradient computations. Besides, we introduce *local-global gradient discrepancy* as $\kappa^2 = \kappa_{\mathbf{A}}^2 + \kappa_{\mathbf{X}}^2$, where $\kappa_{\mathbf{A}}^2 = \max_{p\in[P]}\{\|\nabla\mathcal{L}_p^{\text{local}}(\boldsymbol{\theta}) - \nabla\mathcal{L}_p^{\text{full}}(\boldsymbol{\theta})\|^2\}$ is the maximum difference between the gradient computed on the local machine with and without having access to the global graph structure, which is mainly due to fact that the local machines are oblivious to the full graph information; and $\kappa_{\mathbf{X}}^2 = \max_{p\in[P]}\{\|\nabla\mathcal{L}_p^{\text{full}}(\boldsymbol{\theta}) - \nabla\mathcal{L}(\boldsymbol{\theta})\|^2\}$ is the maximum difference between the gradient computed using the local node and all nodes, which is mainly due to the heterogeneity of the node features on each local machine, and we have $\kappa_{\mathbf{X}}^2 = 0$ if the nodes are i.i.d. sampled to each local machine. Notice that *local-global gradient discrepancy* $\kappa^2$ plays an important role in our theoretical results.

For the convergence analysis, we make the following standard assumptions.

**Assumption 1** *The stochastic gradient on the p-th local machine (with neighbor sampling) has stochastic gradient variance bounded by $\sigma_{var}^2$ and stochastic gradient bias bounded by $\sigma_{bias}^2$, i.e.,*
$$\mathbb{E}[\|\tilde{\nabla}\mathcal{L}_p^{local}(\boldsymbol{\theta};\xi) - \mathbb{E}[\tilde{\nabla}\mathcal{L}_p^{local}(\boldsymbol{\theta};\xi)]\|^2] \leq \sigma_{var}^2, \quad \mathbb{E}[\|\mathbb{E}[\tilde{\nabla}\mathcal{L}_p^{local}(\boldsymbol{\theta};\xi)] - \nabla\mathcal{L}_p^{local}(\boldsymbol{\theta})\|^2] \leq \sigma_{bias}^2.$$

**Assumption 2** *The stochastic gradient for global server correction (with full neighbors) has stochastic gradient variance bounded by $\sigma_{global}^2$, i.e., $\mathbb{E}[\|\tilde{\nabla}\mathcal{L}_p^{full}(\boldsymbol{\theta};\xi) - \nabla\mathcal{L}_p^{full}(\boldsymbol{\theta})]\|^2] \leq \sigma_{global}^2$.*

The existence of stochastic gradient bias and variance in sampling-based GNN training have been studied in (Cong et al., 2020; 2021), where (Cong et al., 2021) further quantify the stochastic gradient bias and variance as a function of the number of GCN layers. In particular, they show that the existence of $\sigma_{bias}^2$ is due to neighbor sampling and non-linear activation, and we have $\sigma_{bias}^2 = 0$ if all neighbors are used or the non-linear activation is removed. The existence of $\sigma_{var}^2$ is because we are sampling mini-batches to compute the stochastic gradient on each local machine during training. As the mini-batch size increases, $\sigma_{var}^2$ will be decreasing, and we have $\sigma_{var}^2 = 0$ when using full-batch.

### 4.2 DISTRIBUTED GNN VIA PARAMETER AVERAGING

In the following, we provide the first convergence analysis on distributed training of GCN. We show that solely periodic averaging of the local machine models and ignoring the global graph structure suffers from an upper bound that is irreducible with the number of training steps. Comparing to the traditional distributed training (e.g., distributed training Convolutional Neural Network for image classification (Dean et al., 2012; Li et al., 2020b)), the key challenges in the distributed GCN training is the two different types of gradient bias: (1) The expectation of the local full-batch gradient is

a biased estimation of the global full-batch gradient, i.e., $\frac{1}{P}\sum_{p=1}^{P}\nabla\mathcal{L}_p^{\text{local}}(\boldsymbol{\theta}) \neq \nabla\mathcal{L}(\boldsymbol{\theta})$. This is because each local machine does not have access to the original input graph and full node feature matrix. Note that the aforementioned equivalence is important for the classifcal distributed training analysis Dean et al. (2012); Yu et al. (2019). (2) The expectation of the local stochastic gradient is a biased estimation of the local full-batch gradient i.e., $\mathbb{E}[\tilde{\nabla}\mathcal{L}_p^{\text{local}}(\boldsymbol{\theta},\xi_p)] \neq \nabla\mathcal{L}_p^{\text{local}}(\boldsymbol{\theta})$. This is because the stochastic gradient on each local machine is computed by using neighbor sampling, which has been studied in (Cong et al., 2021).

**Theorem 1 (Distributed GCN via Parameter Averaging)** *Consider applying model averaging for GNN training under Assumption 1 and 2. If we choose learning rate $\eta = \frac{\sqrt{P}}{\sqrt{T}}$ and the local step size $K \leq \frac{\sqrt{2}T^{1/4}}{8LP^{3/4}}$, then for any $T \geq L^2P$ steps of gradient updates we have*

$$\frac{1}{T}\sum_{t=0}^{T-1}\mathbb{E}[\|\nabla\mathcal{L}(\bar{\boldsymbol{\theta}}^t)\|^2] = \mathcal{O}\left(\frac{1}{\sqrt{PT}}\right) + \mathcal{O}(\kappa^2 + \sigma_{bias}^2).$$

Theorem 1 implies that, by carefully choosing the learning rate $\eta$ and the local step size $K$, the gradient norm computed on the virtual averaged model is bounded by $\mathcal{O}(1/\sqrt{PT})$ after $R = T/K = \mathcal{O}(\frac{P^{3/4}}{T^{3/4}})$ communication rounds, but suffers from an irreducible residual error upper bound $\mathcal{O}(\kappa^2 + \sigma_{\text{bias}}^2)$. In the next section, we show that this residual error can be eliminated by applying server correction.

### 4.3 DISTRIBUTED GCN VIA SERVER CORRECTION

Before proceeding to our result, in order to simplify the presentation, let us first define the notation $G_{\text{global}}^r = \min_{t\in\mathcal{T}_{\text{global}}(r)}\mathbb{E}[\|\nabla\mathcal{L}(\bar{\boldsymbol{\theta}}^t)\|^2]$ and $G_{\text{local}}^r = \min_{t\in\mathcal{T}_{\text{local}}(r)}\mathbb{E}[\|\frac{1}{P}\sum_{p=1}^{P}\nabla\mathcal{L}_p^{\text{local}}(\boldsymbol{\theta}_p^t)\|^2]$ as the minimum gradient computed at the $r$-th round global and local step, where $\mathcal{T}_{\text{global}}(r)$ and $\mathcal{T}_{\text{local}}(r)$ are the number of iteration run after the $r$-th communication round on server and local machine, respectively. Please refer to Eq. 42 in Appendix C.2 for a formal definition.

**Theorem 2** *Consider applying model averaging for GCN training under Assumption 1 and 2. If we choose learning rate $\gamma = \eta = \frac{\sqrt{P}}{\sqrt{T}}$, the local step size $K, \rho$ such that $\sum_{r=1}^{R}K^2\rho^{2r} \leq \frac{RT^{1/2}}{32L^2P^{3/2}}$, and server correction step size $S = \max_{r\in[R]}\left(\frac{\kappa^2 + 2\sigma_{bias}^2}{1-L(\sqrt{P/T})} - G_{local}^r\right)\frac{K\rho^r}{G_{local}^r}$, then for any $T \geq L^2P$ steps of gradient updates we have: $\frac{1}{T}\sum_{t=1}^{T}\mathbb{E}[\|\nabla\mathcal{L}(\bar{\boldsymbol{\theta}}^t)\|^2] = \mathcal{O}\left(\frac{1}{\sqrt{PT}}\right)$.*

Theorem 2 implies that, by carefully choosing the learning rates $\gamma$ and $\eta$, the local step size hyperparameters $K, \rho$, and the number of global correction steps $S$, after $T$ steps ($R$ rounds of communication), employing parameter averaging with *Global Server Correction*, we have the norm of gradient bounded by $\mathcal{O}(1/\sqrt{PT})$, without suffering the residual error that exists in the naive parameter averaging (in Theorem 1). Besides, the server correction step size is proportional to the scale of $\kappa^2$ and local stochastic gradient bias $\sigma_{\text{bias}}^2$. The larger $\kappa^2$ and $\sigma_{\text{bias}}^2$, the more corrections are required to eliminate the residual error. However, in practice, we observe that a very small number of correction steps (e.g., $S = 1$) performs well, which minimizes the computation overhead on the server.

## 5 EXPERIMENTS

**Real-world simulation.** In a real-world distributed setting, the server and local machines are located on different machines, connected through the network (Li et al., 2020a). However, for our experiments, we only have access to a single machine with multiple GPUs. As a result, we simulate a real-world distributed learning scenario, such that each GPU is responsible for the computation of two local machines (8 in total) and the CPU acts as the server. For these reasons, in our evaluations, we opted to report the communication size and number of communication rounds, instead of the wall-clock time, which can show the benefit of distributed training. We argue that these are acceptable measures in real-world scenarios as well since the two main factors in distributed training are initializing connection overhead and bandwidth (Tripathy et al., 2020).

**Baselines.** To illustrate the effectiveness of LLCG, we setup two general synchronized distributed training techniques as the our baseline methods, namely "*Parallel SGD with Parameter Averaging*" (PSGD-PA) and "*Global Graph Sampling*" (GGS), as introduced in Figure 2, where the cut-edges in PSGD-PA are ignored and only the model parameters are transferred, but the cut-edges in GGS are considered and the node features of the cut-edges are transferred to the corresponding machine.

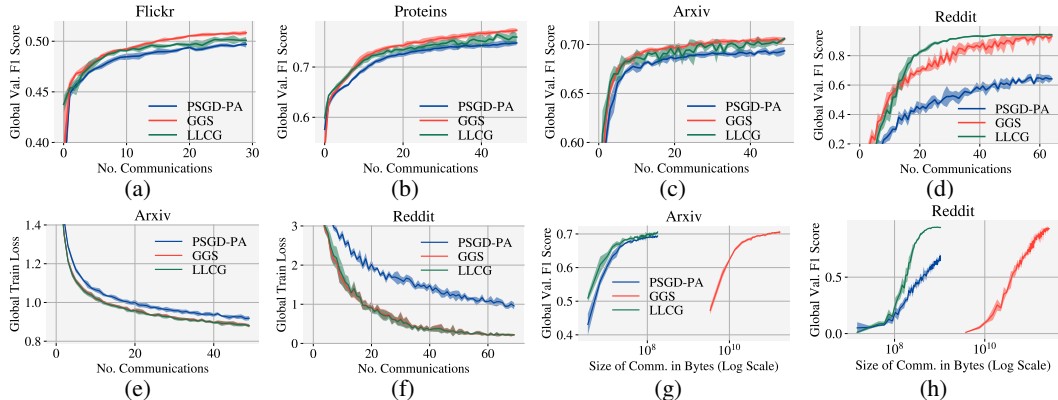

Figure 4: Comparing LLCG against PSGD-PA and GGS on real-world datasets. We show the global validation score in terms of the number of communications in (*a,b,c,d*), the training loss per round of communications in (*e,f*), and the global validation score per bytes of exchanged data in (*g,h*).

Table 1: Comparison of performance and the average Megabytes of node representation/feature communicated per round on various datasets.

| | Method | No. Comm. | GCN / SAGE | | GAT | | APPNP | |
|---|---|---|---|---|---|---|---|---|
| | | | Performance | Avg. MB | Performance | Avg. MB | Performance | Avg. MB |
| Flickr (F1-score) | PSGD-PA | 50 | $49.08_{\pm 0.27}$ | 12.57 | $51.56_{\pm 0.28}$ | 4.24 | $50.81_{\pm 0.48}$ | 8.40 |
| | GGS | | $51.22_{\pm 0.13}$ | 1849.32 | $52.41_{\pm 0.29}$ | 1895.61 | $51.33_{\pm 0.33}$ | 1897.82 |
| | **LLCG** | | $50.38_{\pm 0.20}$ | 12.57 | $52.01_{\pm 0.33}$ | 4.24 | $51.15_{\pm 0.25}$ | 8.40 |
| OGB-Proteins (ROC-AUC) | PSGD-PA | 100 | $72.85_{\pm 0.70}$ | 6.20 | $64.95_{\pm 1.01}$ | 3.14 | $71.10_{\pm 0.79}$ | 7.31 |
| | GGS | | $74.78_{\pm 0.36}$ | 922.42 | $68.11_{\pm 0.60}$ | 912.79 | $71.29_{\pm 0.31}$ | 917.20 |
| | **LLCG** | | $73.92_{\pm 0.45}$ | 6.20 | $67.62_{\pm 0.58}$ | 3.14 | $71.18_{\pm 0.43}$ | 7.31 |
| OGB-Arxiv (F1-score) | PSGD-PA | 100 | $69.43_{\pm 0.21}$ | 3.55 | $69.88_{\pm 0.18}$ | 3.59 | $68.48_{\pm 0.17}$ | 7.71 |
| | GGS | | $70.51_{\pm 0.26}$ | 3391.03 | $70.82_{\pm 0.23}$ | 3396.79 | $69.01_{\pm 0.10}$ | 3394.33 |
| | **LLCG** | | $70.21_{\pm 0.13}$ | 3.55 | $70.58_{\pm 0.37}$ | 3.59 | $68.73_{\pm 0.29}$ | 7.71 |
| Reddit (F1-score) | PSGD-PA | 75 | $71.17_{\pm 1.06}$ | 14.83 | $70.57_{\pm 1.24}$ | 7.48 | $83.48_{\pm 0.81}$ | 11.63 |
| | GGS | | $94.77_{\pm 0.20}$ | 3798.81 | $95.03_{\pm 0.48}$ | 3805.28 | $95.23_{\pm 0.22}$ | 3770.46 |
| | **LLCG** | | $94.67_{\pm 0.15}$ | 14.83 | $94.73_{\pm 0.23}$ | 7.48 | $94.64_{\pm 0.17}$ | 11.63 |

Note that we choose GGS as a reasonable representation for most existing proposals (Md et al., 2021; Zheng et al., 2020; Tripathy et al., 2020) for distributed GNN training, since these methods have very close communication cost and also require a large cluster of machines to truly show their performance improvement. We also use PSGD-PA as a lower bound for communication size, which is widely used in traditional distributed training and similar to the one used in (Angerd et al., 2020; Jiang & Rumi, 2021). However, we did not specifically include these methods in our results since we could not reproduce their results in our settings. Please refer to Appendix A for a detailed description of implementation, hardware specification and link to our source code.

**Datasets and evaluation metric.** We compare LLCG and other baselines on real-world semi-supervised node classification datasets, details of which are summarized in Table 2 in the Appendix. The input graphs are splitted into multiple subgraphs using METIS before training, then the same set of subgraphs are used for all baselines. For training, we use neighborhood sampling (Hamilton et al., 2017) with 10 neighbors sampled per node and $\rho = 1.1$ for LLCG. For a fair comparison, we chose the base local update step $K$ such that LLCG has the same number of local update steps as PSGD-PA. During evaluation, we use full-batch without sampling, and report the performance on the full graph using AUC ROC and F1 Micro as the evaluation metric. Unless otherwise stated, we conduct each experiment five times and report the mean and standard deviation.

## 5.1 PRIMARY RESULTS

In this section, we compare our proposed LLCG algorithm with baselines on four datasets. Due to space limitations we defer the detailed discussion on additional datasets to the Appendix A.4.

**LLCG requires same number of communications.** Figure 4 (a) through 4 (d) illustrate the validation accuracy per communication rounds on four different datasets. We run a fixed number of communication rounds and plot the global validation score (the validation score computed using the full-graph on the server) at the end of each communication step. For PSGD-PA and GGS, the score is calculated on the averaged model, whereas for LLCG the validation is calculated after the correction

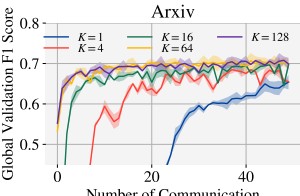
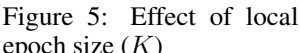
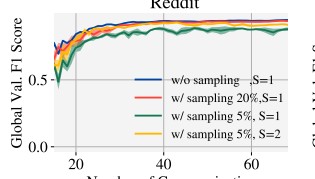
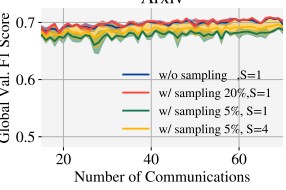

Figure 5: Effect of local epoch size ($K$)

Figure 6: Effect of sampling on local machine and number of correction steps on the server

step. It can be seen that PSGD-PA suffers from performance drop compared to other two methods, due to the residual error we discussed in Section 4, while both GGS and LLCG perform well and can achieve the expected accuracy. Note that the performance drop of PSGD-PA can vary across different datasets; in some cases such as Reddit, PSGD-PA can significantly hurt the accuracy, while on other datasets the gap is smaller. Nevertheless, LLCG can always close the gap between PSGD-PA and GGS with minimal overhead.

**LLCG convergences as fast as GGS.** To represent the effect of communication on the real-time convergence, in Figure 4 (e) and 4 (f), we plot the global training loss (training loss computed on the full-graph on the server) after each communication round. Similar to the accuracy score, the training loss is also computed on the server averaged (and corrected, in case of LLCG) global model. These results clearly indicate that LLCG can improve the convergence over PSGD-PA, while it shows a similar convergence speed to GGS.

**LLCG exchanges data as little as PSGD-PA.** Figure 4 (g) and 4 (h) show the relation between global validation accuracy with the average size (volume) of communication in bytes. As expected, this figure clearly shows the effectiveness of LLCG . On the one hand, LLCG has a similar amount of communication volume as PSGD-PA but can achieve a higher accuracy. On the other hand, LLCG requires significantly less amount of communication volume than GGS to achieve the same accuracy, which leads to slower training time in real world settings.

**LLCG works with various GNN models and aggregations.** We evaluate four popular GNN models, used in recent graph learning literature: GCN Kipf & Welling (2017), SAGE Hamilton et al. (2017), GAT Velickovic et al. (2018) and APPNP Klicpera et al. (2019). In Table 1, we summarize the test score and average communication size (in MB) on different datasets for a fixed number of communication rounds. Note that we only include the results for the aggregation methods (GCN or SAGE) that have higher accuracy for the specific datasets, details of which can be found in Appendix A.2. As shown here, LLCG can consistently improve the test accuracy for all different models compared to PSGD-PA, while the communication size is significantly lower than GGS, since LLCG only needs to exchange the model parameters.

**Effect of local epoch size.** Figure 5 compares the effect of various values of local epoch size $K \in \{1, 4, 16, 64, 128\}$ for fixed $\rho$ and $S$ on the OGB-Arxiv dataset. When using *fully synchronous* with $K = 1$, the model suffers from very slow convergence and needs more communications. Further increasing the $K$ to larger values can speed up the training; however, we found a diminishing return point for $K > 128$ in this dataset and extremely large $K$ in general.

**Effect of sampling in local machines.** In Figure 6, we report the validation scores per round of communication to compare the effect of neighborhood sampling at local machines. We can observe that when the neighborhood sampling size is reasonably large (i.e., $20\%$), the performance is very similar to full neighborhood training. However, reducing the neighbor sampling ratio to $5\%$ could result in a larger local stochastic gradient bias $\sigma_{\text{bias}}^2$, which requires using more correction steps ($S$).

## 6 CONCLUDING REMARKS

In this paper, we propose a novel distributed algorithm for training Graph Neural Networks (GNNs). We theoretically analyze various GNN models and discover that, unlike the traditional deep neural networks, due to inherent data samples dependency in GNNs, naively applying periodic parameter averaging leads to a residual error and current solutions to this issue impose huge communication overheads. Instead, our proposal tackles these problems by applying correction on top of locally learned models, to infuse the global structure of the graph back into the network and avoid any costly communication. In addition, through extensive empirical analysis, we support our theoretical findings and demonstrate that LLCG can achieve high accuracy without additional communication costs.

## ACKNOWLEDGEMENTS

This work was supported in part by CRISP, one of six centers in JUMP, a Semiconductor Research Corporation (SRC) program sponsored by DARPA and NSF grants 1909004, 1714389, 1912495, 1629915, 1629129, 1763681, 2008398.

## REPRODUCIBILITY STATEMENT

We provide a GitHub repository in Appendix A including all code and scripts used in our experimental studies. This repository includes a `README.md` file, explaining how to install and prepare the code and required packages. Detailed instruction on how to use the partitioning scripts is provided for various datasets. In addition, we provide several configuration files (under `scripts/configs`) folder for different hyper-parameters on each individual dataset, and a general script (`scripts/run-config.py`) to run and reproduce the results with these configurations. Details of various models and parameters used in our evaluation studies can also be found in Appendix A.

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
