# OpenReview forum: "Learn Locally, Correct Globally: A Distributed Algorithm for Training Graph Neural Networks"
_ICLR.cc/2022/Conference — ICLR 2022 Poster_

### Official Review · Reviewer_Wu2z · 2021-10-31

**Correctness:** 3
**Technical Novelty And Significance:** 3
**Empirical Novelty And Significance:** 3
**Recommendation:** 6
**Confidence:** 2

**Main Review:**

The paper presents an interesting method for distributed training of GNN which can be helpful when training on very large graphs. The paper is well presented and easy to follow. The theoretical analysis is important in my opinion, as it can provide motivation as to why this method is better than previous ones. Specifically, the importance of having a server correction phase. The empirical part is thorough, providing experiments on several datasets. The proposed training technique is being compared to either PSGD-PA which has worse performance, and GGS which has a much higher communication cost.

I have a couple of concerns regarding the theoretical part:

1) Theorem 1 basically shows an upper bound on the average norm of the gradients throughout T iterations. It is unclear why the “irreducible part” of this bound is really irreducible. I believe that if the authors want to prove that the algorithm from Theorem 2 is better than the one in Theorem 1, then they should show a lower bound in Theorem 1, and make a case that this term is really irreducible.

2) I am not sure if the term discussed in the previous point is really irreducible. Theorem 1 & 2 only shows a bound on the norm of the gradients, averaged over T iterations. If the loss function L is smooth (which may be reasonable), then it would converge eventually to a stationary point. Hence, it is not clear why for a large enough T, this term can be lower bounded by some term independent of T.

3) In Theorems 1 & 2 the authors only show that the loss converges to a stationary point, which may as well be a saddle point or spurious minima. Can the authors elaborate on this? Is there some guarantee on to which point the loss converges to?

As I said, the empirical part is satisfactory in my opinion and shows the benefits of LLCG. It does seem that all the tasks consider node classification, and I wonder how this method would work on edge classification tasks. This may be an important case study since edges can be cut off when splitting the graph into subgraphs.

A few minor issues:

1) The indexing in algorithms 1 & 2 is a bit confusing because there are three loops (global epochs, local machine’s epochs, and server corrections). I think it would be better to use indices for \theta which represent the current global epoch, and either local machine epoch or server correction epoch.

2) Figures 4-7 are very small and hard to read. Due to the page limit constraint, I suggest highlighting fewer figures in the main part and giving the rest of the figures on a larger scale in the appendix.

I note that I’m not an expert in distributed computing, and may have missed some related works or important benchmarks which should have been mentioned.


**Summary Of The Paper:**

This paper proposes a distributed training technique for GNN. This technique includes local computations done in parallel by several machines and a correction phase done by a centralized server. A theoretical analysis is given for this technique, showing that the server correction phase reduces some irreducible error that happens due to splitting the graph and doing a local computation on each subgraph. Several experiments are made on real datasets which show the merits of this technique over previous techniques in terms of performance, communication steps, and size.

**Summary Of The Review:**

I think the optimization technique this paper presents is nice. The empirical results are thorough, showing advantages over previous methods. The theoretical part is a bit unclear. specifically, whether Theorem 1 actually shows a lower bound (i.e. is the second term there irreducible).

---

> ### Author Response · Authors · 2021-11-15
> **Response to Reviewer Wu2z**
>
> > **Q1: Why is the error irreducible? Why for a large enough T, the loss cannot be lower bounded by a term independent of T. The author should show a lower bound in Theorem 1.**
>
> A1: Many thanks for raising this question. The key difficulty in our setting is that, we are not even be able to compute the unbiased estimates of gradients for local objectives. In standard finite sum optimization with Local (S)GD, we can compute an unbiased estimate of local gradient of each local machine *independent* of other machines. Hence, by properly choosing the number of local updates and learning rate we can guarantee the diversity among local objectives vanishes as optimization proceeds (but possibly with slower rate compared to variance term).
>
> However, in our setting the local objectives are connected via cut-edges. In absence of cut-edges, due to partitioning and lack of communication between devices, we are no longer able to compute the  **unbiased** full/stochastic gradient of local objectives. While the diversity among local objectives can vanish as in standard Local SGD discussed above, the bias introduced by cut-edges in computing local stochastic gradients during the course of optimization looks unavoidable. The proposed periodic server correction step aims to compensate for bias introduced by cut-edges and neighbour sampling in computing local stochastic gradients.
>
> Unfortunately, we do not have rigorous explanation of this intuition by providing a lower bound, but in the following we try to elaborate more on this issue based on our analysis and hopefully it will be convincing enough. To answer the question, we start by providing a detailed discussion on why error is irreducible, then explaining why the lower bound is not feasible.

---

> > ### Author Response · Authors · 2021-11-15
> > **Response to Reviewer Wu2z (cont'd)**
> >
> > > **Detailed discussion on why error is irreducible.**
> >
> > To better understand why the error in the upper bound might be irreducible (i.e., the second term in RHS of Theorem 1 is independent of T), let's first recall the key sources that cause gradient diversity
> > $\Big\\| \nabla \mathcal{L}(\bar{\boldsymbol{\theta}}^t) - \frac{1}{P}\sum\_{p=1}^P \mathbb{E}[ \tilde{\nabla} \mathcal{L}\_p^\text{local}(\boldsymbol{\theta}\_p^t;\xi_p^t)] \Big\\|^2 $ (in Eq. 21), in our setting:
> >
> > - The $\kappa^2 $ term, which is the upper bound of $ \frac{1}{P}\sum_{p=1}^P \\| \nabla \mathcal{L}(\bar{\boldsymbol{\theta}}^t) - \nabla \mathcal{L}_p^\text{local}(\bar{\boldsymbol{\theta}}^t) \\|^2$ and is caused by ignoring the cut-edges for local gradient computation;
> > - The $\sigma_\text{bias}^2$ term, which is the upper bound of $\frac{1}{P}\sum_{p=1}^P \left\\| \mathbb{E}[ \tilde{\nabla} \mathcal{L}_p^\text{local}(\bar{\boldsymbol{\theta}}^t;\xi_p^t)] - \nabla \mathcal{L}_p^\text{local}(\boldsymbol{\theta}_p^t) \right\\|^2$ and is caused by using neighbor sampling;
> > - The model divergence term $\frac{1}{P}\sum_{p=1}^P \\| \bar{\boldsymbol{\theta}}^t - \boldsymbol{\theta}_p^t\\|^2$ which is caused by the difference between the model parameters and the virtual averaged model due to periodic averaging. Note that the model divergence term also exists in distributed learning regardless of whether the dataset is graph or not and caused by infrequent synchronization.
> >
> >
> > Now, let's have a closer look at the model divergence term.
> > As shown in Lemma 3 the model divergence term $\frac{1}{P}\sum_{p=1}^P \\| \bar{\boldsymbol{\theta}}^t - \boldsymbol{\theta}_p^t\\|^2$ is further caused by three factors:
> >
> > - The $\sigma_\text{var}^2$ term (in Eq. 64),  which is the upper bound of mini-batch sampling variance $\frac{1}{P}\sum_{p=1}^P \mathbb{E} \Big[ \Big\\|  \sum_{\tau=t_0}^{t-1} \left( \mathbb{E}[\tilde{\nabla} \mathcal{L}_p^\text{local}(\boldsymbol{\theta}_p^{\tau}; \xi_p^{\tau})] - \nabla \mathcal{L}_p^\text{local}(\boldsymbol{\theta}_p^{\tau})  \right) \Big\\|^2 \Big]$
> > - The $\kappa^2 $ term, which is the upper bound of $ \frac{1}{P}\sum_{p=1}^P \\| \nabla \mathcal{L}(\bar{\boldsymbol{\theta}}^t) - \nabla \mathcal{L}_p^\text{local}(\bar{\boldsymbol{\theta}}^t) \\|^2$ and is caused by ignoring the cut-edges for local gradient computation;
> > - The $\sigma_\text{bias}^2$ term, which is the upper bound of $\frac{1}{P}\sum_{p=1}^P \left\\| \mathbb{E}[ \tilde{\nabla} \mathcal{L}_p^\text{local}(\bar{\boldsymbol{\theta}}^t;\xi_p^t)] - \nabla \mathcal{L}_p^\text{local}(\boldsymbol{\theta}_p^t) \right\\|^2$ and is caused by using neighbor sampling.
> >
> > Fortunately, the model divergence term can be controlled by the number of local gradient update steps and learning rate, which is reducing with respect to the number of total training steps $T$ and the number of local machine $P$, and it leads to the first term in our upper bound in Theorem 1, i.e., $\mathcal{O}(\frac{\sigma_\text{var}^2 + \sigma_\text{bias}^2 + \kappa^2}{\sqrt{PT}}) = \mathcal{O}(\frac{1}{\sqrt{PT}})$.
> > However, unfortunately, $\kappa^2$ and $\sigma^2_\text{bias}$ in gradient diversity are isolated from the model diversity part, therefore are irreducible and results in $\mathcal{O}(\sigma_\text{bias}^2 + \kappa^2)$ in the second term in our upper bound in Theorem 1. Please refer to Figure 12 for an illustration.
> >
> > Intuitively, these theoretical results make sense. During training, we are minimizing the loss without cut-edges $\frac{1}{P}\sum_{p=1}^P \mathcal{L}^\text{local}(\boldsymbol{\theta})$, which is a different objective to the loss defined on the full-graph by taking the cut-edges into consideration $\mathcal{L}(\boldsymbol{\theta})$. Therefore, solely by adding the number of training iterations, we cannot guarantee a small gradient of $\mathcal{L}(\boldsymbol{\theta})$ by minimizing $\frac{1}{P}\sum_{p=1}^P \mathcal{L}^\text{local}(\boldsymbol{\theta})$.

---

> > > ### Author Response · Authors · 2021-11-15
> > > **Response to Reviewer Wu2z (cont'd)**
> > >
> > > > **Why lower bound is not feasible?**
> > >
> > > Although the lower bound for the gradient norm seems like a promising way to explain the irreducibility, in the optimization, a lower bound is obtained and used in a different way, which might be different from what the reviewer is expecting.
> > > The lower bound is obtained by coming up with a worst special case (cannot directly apply to any specific problem), then it is used to be compared with the upper bound to show the "tightness" of the upper bound.
> > >
> > > For example, in the Appendix D.4 of [1], they provide a lower bound by constructing a hard problem $f(x) = f\_1(x) + f\_2(x)$, where $f\_1(x) = \mu x^2 + Gx$ and $f\_2(x) = -G x$. This problem is considered as a worst special case because both their optimal solution are not all equal (i.e.,  $\arg\min\_x f(x) = \arg\min\_x f\_2(x) \neq \arg\min\_x f\_1(x) $, where $\arg\min\_x f(x) = 0$, $\arg\min\_x f\_1(x) = -\frac{G}{2u}$, and $\arg\min\_x f\_2(x) = 0$) and their gradiensts are different (i.e., $ f^\prime(x) = 2\mu x$, $ f\_1^\prime(x) = 2\mu x + G$, and $ f\_2^\prime(x) = - G$). Then, they provide some worse case analysis on top of this special case. Such type of analysis is not adaptable to ours due to impossibility of computing unbiased local stochastic gradients which goes beyond vanilla finite-sum minimization. To the best of our knowledge, even providing the lower bound for finite sum non-convex setting in a distributed learning setting is still an open problem. We believe this is an interesting but challenging problem and we will mention it as a future work. Once again many thanks for raising this important question.
> > >
> > >
> > > [1] [SCAFFOLD: Stochastic Controlled Averaging for Federated Learning](https://arxiv.org/abs/1910.06378)
> > >
> > >
> > > > **Q2: Is there some guarantee on to which point the loss converges to? e.g., saddle point or spurious minima.**
> > >
> > > A3: Thank you for your question. Unfortunately, existing understanding of the non-convex optimization landscape is still limited. Existing results on the optimization landscape of non-convex problems are either on matrix factorization problems or linear neural networks (without activation function) with some assumptions on the weight matrices. Current theoretical understanding on non-linear neural networks (e.g., neural networks with activation function) is still an open problem. We are sorry that we cannot provide a guarantee on which point the model is converging to. However, note that any result for (linear) neural network also hold for GNN by either setting adjacency as identity or by thinking the graph convolutions are applied to node features before feeding them into a neural network. Besides, it is known that the randomness in stochastic gradients can help neural networks get rid of these saddle points and converge to some local minima. And indeed, our experiments show that the performance of our model can match the state-of-the-art performance achieved by training locally on a single machine.
> > >
> > > > **Experiment suggestions:**
> > >
> > > Thanks, we greatly appreciate the careful reading of papers and thoughtful suggestions.
> > > During the rebuttal period, we also tried to apply PSGD-PA to *OGB-Citation2* dataset based on [this implementation](https://github.com/snap-stanford/ogb/blob/master/examples/linkproppred/citation2/sampler.py) for link prediction (binary classification on whether an edge exists) on 16 local machines, simulated in distributed training environment. However, we observe that there is no obvious performance gap between fully sync and PSGD-PA. This might due to the fact that link prediction is too simple or the negative sampling strategies in the link prediction tasks make the issue of ignoring the cut-edges negligible.  Since the design of edge classification and link prediction task are different from the node classification task, we decide to make it as a potential future direction from the node classification problem discussed in this paper.

---

> > ### Comment · Reviewer_Wu2z · 2021-11-28
> > **Re: Response**
> >
> > I thank the authors for their response.
> > My two main concerns remain, but I will maintain my score because I still think that the optimization technique is nice and the empirical part is good.
> >
> > I will elaborate more on my two concerns:
> >
> > 1) Based on Theorems 1 & 2, to my understanding, it is not possible to claim that with server correction has a faster convergence rate than without server correction. Basically, Theorem 1 shows that without server correction the convergence is in time $O(t_1) + O(t_2)$, while Theorem 2 shows that with server correction the convergence is in time $O(t_1)$. Based only on those results, it is not clear which algorithm is faster, and a lower bound is required. It may be an open problem to achieve this lower bound, which is why I don't expect this paper to show such proof, but still, the authors can't claim formally that one algorithm performs better than the other one.
> >
> > 2) Regarding the irreducible term. I understand that the authors have some intuition on why this term is really irreducible, but there is still something missing regarding the optimization process. The l.h.s of the displayed equation in Theorem 1 is something of the form $1/T \sum_{i=1}^{T-1} \mathbb{E}[||\nabla L(\theta_t)||]$. Focusing just on this term for the moment, if the function $L$ is smooth then GD (or SGD with bounded noise) should converge to a stationary point of $L$, which means that this term should go to zero as $T\rightarrow \infty$. Even without the context of GNN and that a distributed algorithm is used, GD should converge just by using the smoothness of $L$. If the authors claim that the term is really irreducible, then GD will not converge to a stationary point, even if we run the algorithm infinitely many iterations. This seems like a very weird conclusion, is it because the function $L$ is not smooth? Or is it possible that without server correction, the algorithm will just never converge?

---

> > > ### Author Response · Authors · 2021-11-29
> > > **Additional Response to Reviewer Wu2z**
> > >
> > > Many thanks for taking the time and checking our responses. We agree with the reviewer that claiming about the irreducibility of the error might be too strong.
> > >
> > > 1. With regards to the improvement in upper bound obtained by Theorem 2 compared to Theorem 1, we note that $\mathcal{O}(t\_2) = \mathcal{O}(\kappa^2 + \sigma^2_\text{bias})$ term in Theorem 1 dominates the first term $\mathcal{O}(t\_1)$ as long as $T$ is large enough. This $\mathcal{O}(t\_2)$ error is independent of the number of iterations the algorithm proceeds and corresponds to residual error (bias) caused by cut-edges and neighbor sampling which is also observed in our experiments. Theorem 2 indeed shows that the server correction steps can correct this bias and alleviate $\mathcal{O}(t\_2)$ term which entails a rate that would have been obtained by an ideal distributed training without biased gradients.
> > >
> > >
> > > Per your comment, we will try to be more careful in using the term "irreducible error". As we do not have any lower bound, we will update "irreducible error" to "upper bound that is irreducible with the number of training steps" and provide a discussion on why lower bound is still an open problem and is hard to achieve in our next draft.
> > >
> > >
> > > 2. Thanks for your great comment. We totally agree that the LHS vanishes and the algorithm converges to a stationary point provided that we run the algorithm for enough number of iterations. This holds in general when we have access to an oracle that provides noisy *unbiased* stochastic gradients of objective. However, in our setting, we only have access to an oracle that provides noisy *biased* stochastic gradients. This requires a bias correction schema (server updates) to guarantees convergence to a stationary point to alleviate the negative effect of bias.
> > >
> > > Once again, many thanks for your valuable time and great comments. We will gladly incorporate all these comments in subsequent version of our paper.

---

### Official Review · Reviewer_qLfR · 2021-10-31

**Correctness:** 4
**Technical Novelty And Significance:** 3
**Empirical Novelty And Significance:** 3
**Recommendation:** 6
**Confidence:** 3

**Main Review:**

This paper deals with the problem of distributed training of GNNs. Existing methods are either communication-intensive (sampling) or do not achieve good performance (averaging).

The authors propose a novel method, dubbed "LLCG: Learn Locally Correct Globally". Essentially, this method captures the idea of transmitting only local averages but adds a centralized step on the server to account for global structural information lost in the subgraph partition.

The authors further provide theoretical convergence guarantees. They both show that just averaging leads to an insurmountable residual error that explains the poor performance of averaging methods, as well as prove that this residual error disappears when adding the global correction step.

The paper is fairly well written and the proposed result, albeit simple, is powerful. The theoretical guarantees provided round up the good work. More importantly, the topic is timely and very relevant, and I, therefore, recommend acceptance in the conference, provided they add standard deviation results to Table 1.

Some questions that could further enhance the discussion are as follows:

1) What would happen if all labeled nodes used for training end up in a single server. How long would it take to converge? Would it be affected?

2) Are there any best ways of partitioning the graphs into the servers? How does partitioning affect convergence?

Minor comments:

Beginning of Section 4.3: where it reads 'before processing' it should read 'before proceeding'

The paragraph 'datasets and evaluation metric' refers to six semi-supervised classification datasets and points to Table 2. However, the results are for four datasets and referenced in Table 1, not table 2. I understand that there are more experiments on the supplementary material, but refer to table 1 and just four datasets in this section since that are what is being shown in the main matter of the paper.

Fig 4 f, I understand this should be Proteins, and not Reddit (Reddit is in Fig 4h)

**Summary Of The Paper:**

This paper deals with the problem of distributed training of GNNs. Existing methods are either communication-intensive (sampling) or do not achieve good performance (averaging).

The authors propose a novel method, dubbed "LLCG: Learn Locally Correct Globally". Essentially, this method captures the idea of transmitting only local averages but adds a centralized step on the server to account for global structural information lost in the subgraph partition.

The authors further provide theoretical convergence guarantees. They both show that just averaging leads to an insurmountable residual error that explains the poor performance of averaging methods, as well as prove that this residual error disappears when adding the global correction step.

**Summary Of The Review:**

The paper is fairly well written and the proposed result, albeit simple, is powerful. The theoretical guarantees provided round up the good work. More importantly, the topic is timely and very relevant, and I, therefore, recommend acceptance in the conference, provided they add standard deviation results to Table 1.

My assigned score is 7.

---

> ### Author Response · Authors · 2021-11-15
> **Response to Reviewer qLfR**
>
> > **Q1: What would happen if all labeled nodes used for training end up in a single server. How long would it take to converge? Would it be affected?**
>
> A1: This is indeed an interesting question. We need to discuss two different possibilities. Let suppose we train 2-layer GNN for the ease of presentation.
>
> - (All labeled nodes are located on one machine, all 2-hop neighbors are also on the same machine) Under such a setting, the result won’t be affected because the $\kappa_A=0$ and $\kappa_X=0$ (notation introduced in Section 4.1). This is intuitively correct since all nodes that are used for training are available.
> - (All labeled nodes are located on one machine, some of the 2-hop neighbors are on the other machines) Under such a setting, we have $\kappa_A \neq 0$ but $\kappa_X=0$. This can be seen as training of the original graph on a single machine, but with some of the nodes/edges being removed.
>
> > **Q2: Are there any best ways of partitioning the graphs into the servers? How does partitioning affect convergence?**
>
> A2: Indeed, partitioning will affect the performance of distributed training of GNNs. According to our observation (as well as other papers on distributed GNN training, discussed in the related work section), partitioning the graph with minimum cut-edges using METIS is the most promising option. Other partition strategies, e.g., random partition, will result in too many cut-edges, therefore make the experiment result on PSGD-PA even worse.
>
> > **Minor comments:**
>
> We truly appreciate your careful reading of our paper and will make sure to fix the typos. We also included the standard deviation in Table 1 in the updated draft. Also note that Figure 4 (e and f) and 4 (g and h) are for Arxiv and Reddit only.

---

> > ### Comment · Reviewer_qLfR · 2021-11-19
> > **Concerns Addressed**
> >
> > I would like to thank the authors for satisfactorily addressing my concerns.

---

> > > ### Author Response · Authors · 2021-11-29
> > > **Response to Reviewer qLfR**
> > >
> > > Many thanks for taking the time and checking our responses. We are glad that your concerns were resolved. We  appreciate your constructive feedback and will gladly incorporate them in the updated draft.

---

### Official Review · Reviewer_5Wg7 · 2021-11-01

**Correctness:** 3
**Technical Novelty And Significance:** 3
**Empirical Novelty And Significance:** 3
**Recommendation:** 6
**Confidence:** 3

**Main Review:**

The paper is well-written and is enjoyable to read. The problem is clearly defined, and the solution steps are thoroughly explained. I do have some concerns, which I hope the authors will address:
- It would help if the GGS algorithm is explained (at least at a high level) in the paper since it figures some prominently in all the experiments.
- In equation 1, the h update equation seems independent of input features x.
- Since server correction is claimed to be crucial for performance, it would help if the authors can do an ablation study without server correction. I wonder if this is the same as PSGD-PA since LLCG uses exponential increasing round lengths whereas PSGD-PA does not.
- Are the graph partitions at the local machines the same for PSGD-PA and LLCG in the experiments?
- In the server correct step, does it help for the server to carefully choose the minibatch to include more of the cut-edges (i.e., edges that are not considered by the local machines)?
- What is T_global(r) and T_local(r) in Section 4.3?
- In Theorem 2, if T = \sum_{r=1}^R K\rho^r, how can \sum_{r=1}^R K^2 \rho^{2r} be less than O(\sqrt{T})?
- In figure 4 (and others), the performance is plotted as a function of number of communication rounds. How does performance fare as a function of number of local training steps?


**Summary Of The Paper:**

The paper considers the problem of distributed training for graph learning tasks, under a setting where data privacy is significant for each individual machine and communication to/from a central parameter server is expensive. To preserve privacy each machine has only access to a distinct partition of the overall graph. The central server has access to the full graph. In the LLCG algorithm that the paper proposes, each machine trains on its local graph partition for some time before sending the parameters to the server. The server averages the received parameter, but additionally also does its own training using the full graph available to it. Theoretically the authors show that the proposed method avoids an error gap in the gradient norm that would exist if server correction is not performed. Experimental results show the proposed scheme performing similarly to GGS albeit with much lower communication costs.

**Summary Of The Review:**

Well-written paper that considers an important problem and presents a simple, effective solution. There are some concerns however which I hope the authors will address.

---

> ### Author Response · Authors · 2021-11-15
> **Response to Reviewer 5Wg7**
>
> > **Q1: Explain GGS at a high level.**
>
> A1: Thank you for mentioning this. We introduce GGS in the caption of Figure 2 and we refer to this figure when we discuss GGS. However, during revision, we will place the definition in a more conspicuous position.
>
> > **Q2: The h update equation seems independent of input features x:**
>
> A2: We appreciate your careful reading and pointing this out. In fact, the 0-th layer $\mathbf{h}_i$ is the input $\mathbf{x}_i$, i.e., $\mathbf{h}_i^{(0)}=\mathbf{x}_i$. We will clarify during refinement.
>
> > **Q3: Experiments: without server correction but with exponential local steps on PSGD-PA.**
>
> A3: Thank you for your suggestions. The design of exponential local steps is mainly used to balance the computation and communication cost, and provide a more flexible local update schema. To verify whether exponential local steps alone can improve the accuracy, we conduct an experiment by comparing the PSGD-PA with exponentially increasing local steps sizes (similar to our proposed LLCG) with LLCG. The results can be found from [Arxiv](https://anonymous.4open.science/r/llcg-2733/figures/psgd-exp-arxiv.pdf) and [Reddit](https://anonymous.4open.science/r/llcg-2733/figures/psgd-exp-reddit.pdf). As shown in the figures, PSGD-PA with exponentially local steps still suffers from performance drop compared to LLCG, therefore exponential local steps alone cannot solve the issue.
>
> > **Q4: Are the graph partitions at the local machines the same for PSGD-PA and LLCG in the experiments?**
>
> A4: Yes, they all use the same partitioning package METIS. Given the original graph, we first partition the graph into subgraphs using METIS, then LLCG and all baseline methods are applied to the same subgraphs for a fair comparison. We will emphasize (using bold face or as a footnote) our setting during revision.
>
> > **Q5: In the server correct step, does it help for the server to carefully choose the minibatch to include more of the cut-edges (i.e., edges that are not considered by the local machines)?**
>
> A5: Thank you for the question. This is indeed an interesting idea and we agree that sampling more cut-edges for server correction seems more favorable at the first glance.
> We provide the results of such experiments by comparing uniform sampling with sampling more cut-edges (i.e., max. cut edges mini-batch in the figure) [Arxiv](https://anonymous.4open.science/r/llcg-2733/figures/srv-mb-arxiv.pdf) and [Reddit](https://anonymous.4open.science/r/llcg-2733/figures/srv-mb-reddit.pdf).
> According to our experimental studies, we discovered that sampling more cut-edges does not make significant improvement when comparing to selecting mini-batchs using uniform sampling.
> This potentially due to sampling more cut-edges make the gradient of server correction step biased, but we only need an unbiased gradient for server correction to compensate for the residual error caused by ignoring cut-edges in computing local gradients.
>
>
> > **Q6: What is $T_\text{global}(r)$ and $T_\text{local}(r)$ in Section 4.3?**
>
> A6: We apologize for the confusion. $T\_\text{global}(r)$ and $T\_\text{local}(r)$ are defined in Eq. 42 in the appendix.
> $T\_\text{local}(r)$ is the number of iteration that a local machine $T\_\text{global}(r)$ is the number of iteration servers run.
> We will introduce this notation in the main text and refer to Eq. 42 in our refined submission.
>
> > **Q7: In Theorem 2, if $T = \sum_{r=1}^R K\rho^r$, how can $\sum_{r=1}^R K^2 \rho^{2r}$ be less than $O(\sqrt{T})$?**
>
> A7: Thanks again for checking carefully. Indeed, it is non-trivial to see how this inequality holds because the relation between $T$ and $L^2 P$  also need to be considered to check why the inequality holds. We suggest the reviewer check Eq. 57-59. However, we can do a quick sanity check by setting $\rho=1$. Then, using $T>L^2P$, we can immediately get $K<\frac{T^{1/4}}{L P^{3/4}}$, which recover the result of PSGD-PA in Theorem 1.
>
> > **Q8: Comparing the number of communication rounds fair? How about the number of local training steps?**
>
> A8: Thank you for asking. The comparison is fair because all baseline methods and ours are running the same number of local update steps $T$ and the communication steps. For example, compared to PSGD-PA, LLCG is using smaller local update steps at the beginning than PSGD-PA, and slowly increasing the number of local update steps after every communication step, such that the total number of local update steps of LLCG is the same as PSGD-PA.

---

> > ### Comment · Reviewer_5Wg7 · 2021-11-27
> > **Response to authors**
> >
> > Thanks for satisfactorily addressing the concerns that were raised. I have adjusted my score to reflect this.

---

> > > ### Author Response · Authors · 2021-11-29
> > > **Response to Reviewer 5Wg7**
> > >
> > > We greatly appreciate your time and consideration. We are glad that you found our response satisfactory and we will make sure sure to incorporate your constructive feedback in the subsequent version.

---

### Official Review · Reviewer_wvQK · 2021-11-03

**Correctness:** 3
**Technical Novelty And Significance:** 2
**Empirical Novelty And Significance:** Not applicable
**Recommendation:** 5
**Confidence:** 3

**Main Review:**

Major comments:

I am mainly concerned about the novelty of this submission. Local SGD [1] is not very novel in distributional optimization. This submission inserts an additional update on the server to eliminate the sampling error. In my opinion, It is difficult to say that this method was designed specifically for training GNN since one can extend this procedure to any dataset. (This paper argues that training GNN needs this step due to the residual error,  which I will comment on later.) The theoretical results are also not novel. The analysis follows the local sgd method. The only additional condition is the sampling error, which doesn't provide any new insight.

The motivation regarding the proposed method is not well justified. The main contribution, if any novelty, is the full neighbor updates on the server. Why do local machines use sampled neighbors while the server uses full neighbors? The motivation discussed in the introduction mentioned local devices lack the global graph structure. In my opinion, this setting is more of partitioned graphs rather than sampling. This problem also partially involves my above comment. Since the residual error seems to come from the sampling, the improvement from the update on the server is not exciting.

The experimental results show some improvement, particularly for Reddit.  Besides numbers of communication, the running time is also necessary since the server requires additional updates. Table 1 and Figure 4 mentioned communication cost, which is not clearly defined anywhere in this submission.

[1] Stich, Sebastian U. ``Local SGD converges fast and communicates little." arXiv preprint arXiv:1805.09767 (2018).

Minor comments:

The experiments use neighbor sampling. It is interesting to see the impact of different samplers, e.g. importance sampling.

Typo and grammar mistakes, e.g., ``we theoretically analysis the convergence of our proposal LLCG'' -- > analyze. ``During the correction, the server first construct a mini-batch...'' -- > constructs. ``however, due to space limitations we differ the detailed discussion on these datasets to the Appendix'' -- > perhaps refer? Etc.


**Summary Of The Paper:**

Training GNNs is challenging due to high communication costs or large memory overheads. This paper proposes a communication-efficient distributed GNN training technique named Learn Locally, Correct Globally (LLCG) to periodically model averaging on the server using locally trained models.  It also applies global server corrections to refine the locally learned models and solve the irreducible performance degradation caused by ignoring node dependency. This paper provides the convergence analysis and shows the proposed method can address the residual error. The experimental results show significant improvement compared to existing methods.

**Summary Of The Review:**

This paper applies local SGD to GNN training. The idea is not very novel. The experimental results are promising, but some important points need clarification.

---

> ### Author Response · Authors · 2021-11-15
> **Response to Reviewer wvQK**
>
> Thanks for taking the time to review our paper and we highly appreciate your thoughtful suggestions. In the following we summarize and address your main concerns:
>
> > **Q1: Novelty concerns: (Algorithm perspective) This paper only inserts an additional update to the LocalSGD algorithm. Without sampling, it doesn’t need server update; (Theory perspective) The analysis follows the LocalSGD’s method, and the only additional condition is the sampling error.**
>
> **A1:**
> We appreciate your comments, however, we do not fully agree with the reviewer as we realized there might be a misunderstanding about the two types of errors in applying LocalSGD (i.e., PSGD-PA) to graph structured data and potential overlooking of the technical challenges of analyzing the convergence rate in this setting. Hence, before proceeding to this particular question, we would like to first recall and emphasize the key challenges and observations we discussed in this paper.
>
> **Challenges:** As we first mentioned in the first paragraph of Section 4.2, there are two types of error in applying LocalSGD for training GNNs:
>
> - In the analysis of Local SGD (both in IID and non-IID settings), the key assumption is that the local devices are able to compute an **unbaised** estimate of their local gradients. However, in our setting due to absence of cut-edges, this does not hold anymore. In other words, due to ignoring cut-edges, the expectation of the local full-batch gradient is a biased estimation of the global full-batch gradient, i.e., $\frac{1}{P}\sum_{p=1}^P \nabla \mathcal{L}(\boldsymbol{\theta}_p^\text{local}) \neq \nabla \mathcal{L}(\boldsymbol{\theta})$ (this error is illustrated in Figure 3(b) and  denoted by $\kappa^2$ in our analysis). We note that this is true even without intermediate node sampling to compute local stochastic gradients.
> - When we do neighbour sampling, the bias issue becomes more severe as we need to overcome the additional bias caused by sampling. Specifically, due to applying neighbor sampling, the expectation of the local stochastic gradient becomes a biased estimation of the local full-gradient, i.e., $\mathbb{E}[\tilde{\nabla} \mathcal{L}_p^\text{local}(\boldsymbol{\theta}, \xi)] \neq \nabla \mathcal{L}_p^\text{local}(\boldsymbol{\theta})$ as shown in Figure 3(c) and denoted by $\sigma_\text{bias}^2$ in our analysis. Unlike Local SGD where it is assumed that local devices are able to compute an unbiased stochastic estimate of the gradient of the global objective. As a result, bounding the deviation between (virtual) averaged model and local models becomes more challenging due to periodic averaging, and biases caused by lack of cut-edges and neighbor sampling. This is in contrast to standard analysis in Local SGD where we only need to consider the effect of periodic averaging. From this standpoint, our analysis can be considered as generalization of existing analysis of Local SGD to the setting with biased local gradients and it would be interesting by its own.
>
> - To compensate the bias issues raised by absence of cut-edges and neighbour sampling, we propose a global correction step. While the idea looks intuitive and straightforward,  the analysis of convergence rate is involved.
>
> *We note that when using mini-batch with all-neighbors or full-batch with all-neighbors, $\sigma_\text{bias}^2$ disappears but $\kappa^2$ still exists. Therefore, the claim "the only addition is sampling error (sampling error is caused by $\sigma_\text{bias}$" is not accurate.*
>
>
> **Answer to question:** In the following, we answer the main concern of the reviewer:
>
> - (Algorithm novelty) Indeed, our algorithm is developed on top of the LocalSGD algorithm (i.e., PSGD-PA) to solve the distributed training challenges specialized on graph. Our additional update steps are motivated by our analysis on the two types of the gradient bias (i.e., the aforementioned $\kappa$ and $\sigma_\text{bias}$) of generalizing the LocalSGD algorithm to graph structured data. Please note that even without sampling, the $\kappa^2$ still exists, such that the server correction step is still required.
> - (Theory novelty) Existing analysis on Local SGD requires assumptions that the expectation of the local full-batch gradient is an unbiased estimation of the global full-gradient (i.e., assume $\kappa^2$ = 0), however, this is never true when applying LocalSGD on graphs due to the missing cut-edges.  As a result, an involved analysis is required when the unbiasedness assumption does not hold and we employ correction steps, and we would like to emphasize that our results cannot be obtained by direct application of existing analysis of Local SGD.

---

> > ### Author Response · Authors · 2021-11-15
> > **Response to Reviewer wvQK (cont'd)**
> >
> > > **Q2: Experiments: (1) Besides numbers of communication, the running time is also necessary since the server requires additional updates (2) Table 1 and Figure 4 mentioned communication cost, which is not clearly defined anywhere in this submission.**
> >
> > **A2:** (1) We apologies that we do not have the required hardware to compute the exact time in the real distributed environment. However, as a remedy, we provide additional experiments on OGB-Products [(result-link)](https://anonymous.4open.science/r/llcg-2733/figures/ogb-products-acc-vs-comm.pdf) and OGB-MAG240M [(result-link)](https://anonymous.4open.science/r/llcg-2733/figures/ogb-mag240m-acc-vs-comm.pdf) datasets on the synthetic/simulated environment with 16 local machines. We provide computation time for both the local and global steps. For the detailed setup and discussion please refer to the Appendix A.5 in the refined version of the submission.
> > (2) Sorry for the confusion. *Communication per round (in MB)* is referring to the communication cost measured by the number Megabytes of the node hidden representations,node features or model parameters needed to communicate per round, where the hidden representations and features are stored as the `torch.float32` type.

---

### Author Response · Authors · 2021-11-18
**Rebuttal Changes**

We appreciate the constructive comments and suggestions from all reviewers. We applied these clarifications and suggestions (as much as feasible) in our updated draft and highlighted all of these changes in red.
Notably, we made the following changes:

- Section 4.2: Added clarification on the number of local and global iterations
- Section 4.3: Added definition for $G_\text{global}^r$ and $G_\text{local}^r$
- Section 5: Added more detail about the GGS method. Added the standard deviations to Table 1. Included the partitioning details.
- Section A.3: Added new experiments on the effect of minibatch sampling algorithm for the correction steps.
- Section A.5: Provided more experiments on the effectiveness of LLCG on the large-scale setting under a simulated environment.
- Section B.3: Provided a detailed discussion on why the second term on the RHS of Theorem 1 is irreducible.

---

### Author Response · Authors · 2021-11-20
**A kind reminder**

We want to thank all the reviewers, again, for the constructive comments and thoughtful reviews. As a follow-up, we would like to kindly remind the reviewers that the discussion period is ending soon. We want to make sure that the reviewers found our responses coherent and convincing. And we would be more than happy to provide more information/clarification.

---

### Decision · Program_Chairs · 2022-01-20

**Decision:**

Accept (Poster)

**Comment:**

Dear Authors,

The paper was received nicely and discussed during the rebuttal period. The current discussions mostly lie on the acceptance side.

Some prons of the paper include:

- Timely topic: This paper deals with the problem of distributed training of GNNs.
- New algorithm: this method captures the idea of transmitting only local averages but adds a centralized step on the server to account for global structural information lost in the subgraph partition.
- Theory: The authors further provide theoretical convergence guarantees.
- Clarity: The paper is fairly well written and the proposed result is simple and powerful.

The current consensus is that the paper deserves publication.

Best AC